# A virtual intervention to support educator well-being and students' mental health in conflict-affected Ukraine: A non-randomized controlled trial

## Research Article

educators; mental health; school intervention; conflict; psychological support

**Corresponding author:**
Tara Powell;
Email: tlpowell@illinois.edu

Tara Powell[1] ⬤, Natalia Portnytska[2], Iryna Tychyna[2], Olha Savychenko[2], Oksana Makarenko[3], Tetiana Shyriaieva[4], Kate Cherniavska[3], Jenna Muller[1] and Rebecca Carney[5]

[1]School of Social Work, University of Illinois, Urbana, IL, USA; [2]Department of Social and Applied Psychology, Zhytomyr Ivan Franko State University, Zhytomyr Oblast, Ukraine; [3]Smart Osvita, Kyiv, Ukraine; [4]School of Psychology and Counselling, The Open University (UK), Milton Keynes, United Kingdom and [5]Americares, Stamford, CT, USA

## Abstract

The war in Ukraine has caused widespread destruction, displacement, and distress. Educators are among those significantly affected by the conflict, facing the dual burden of educating youth directly impacted by the conflict while simultaneously dealing with their own psychological stress. This study evaluated the Psychosocial Support for Educators (PSE) program, a virtual intervention designed to improve Ukrainian educators' mental health, knowledge, and readiness to support students. A non-randomized control trial included 881 educators from three Ukrainian regions, with 572 participants in the PSE group and 309 in the control group. Surveys assessed psychosocial support knowledge, readiness, and mental health at baseline, post-intervention, and one-month follow-up. Linear mixed model analyses revealed significant improvements in the PSE group across all measures. PSE participants reported greater increases in knowledge ($t = 2.97$, $p = .003$, $d = .38$) and readiness to support students ($t = 6.63$, $p < .001$, $d = .85$), with sustained gains at follow-up. They also reported greater reductions in stress ($t = 2.70$, $p < .01$, $d = .35$), anxiety ($t = 3.20$, $p = .001$, $d = .41$), and depression ($t = 2.00$, $p < .05$, $d = .26$) compared to the control group.

The findings demonstrate that PSE can effectively enhance educators' mental health and their ability to support students in conflict-affected settings, underscoring the importance of accessible, tailored mental health interventions for educators in crisis zones.

## Impact statement

Ukrainian educators are among those significantly affected by the ongoing conflict with Russia, bearing the dual burden of educating youth directly impacted by the conflict while simultaneously managing their own psychological stress. In this study we tested the effectiveness of the Psychosocial Support for Educators (PSE) program in addressing the mental health needs of Ukrainian educators and enhancing their capacity to support students during the Russian invasion in Ukraine. By leveraging a virtual intervention tailored to the unique challenges of educators in war-affected settings, the PSE intervention significantly improved participants' psychosocial knowledge, readiness to assist students, and mental health outcomes. These findings underscore the critical importance of accessible, context-sensitive mental health interventions in equipping educators to navigate crises while fostering resilience in their communities.



## Introduction

The conflict between Ukraine and Russia, which began in 2014 and dramatically escalated in 2022, has been declared the largest military conflict in Europe since World War II, resulting in widespread destruction, significant civilian casualties, and mass displacement within Ukraine and across Europe (Bin-Nashwan *et al.*, 2022). More than 14 million Ukrainians have been displaced from their homes, and the invasion has resulted in nearly 7,000 civilian deaths (Human Rights Watch, 2023). The war has also disrupted education, health care, and other essential services, exacerbating the hardships faced by civilians (Mandragelya, 2022; Barten *et al.*, 2023).

Educators (i.e. teachers) are among those who are deeply impacted by conflict, as they play the dual role of attending to the emotional needs of war-affected children while simultaneously

coping with their own uncertainties and loss (Slone *et al.*, 2021). Schools are frequently closed or repurposed for military use, leaving teachers without a workplace and students without a safe learning environment (Muthanna *et al.*, 2022). This disruption not only halts the educational process but can also exacerbate the psychological stress experienced by educators, who are navigating the danger and instability of living in a conflict zone (Sharifian and Kennedy, 2019). Teachers may face a range of traumatic experiences such as displacement, loss of loved ones and threats to personal safety, which can hinder their ability to perform their duties effectively (Sharifian and Kennedy, 2019). Given the significant emotional burden and the critical role educators play in supporting students during this crisis, educator-focused psychosocial interventions are urgently needed to help teachers manage their own mental health while fostering resilience in their students. Thus, the goal of the present study is to examine the effectiveness of a psychosocial support intervention for educators in Ukraine during the ongoing conflict.

## Mental health in Ukraine

War and conflict can have profound physical and psychological implications for those affected (Kurapov *et al.*, 2023). The ongoing conflict has deeply impacted the well-being of Ukrainians, resulting in widespread trauma, depression, anxiety, and grief as they cope with displacement, the loss of loved ones, and the ongoing threat of violence (Ayer *et al.*, 2017; Khan and Altalbe, 2023). Research has found that up to 62% of Ukrainian adults have reported high anxiety and 58% depression symptoms since the start of the war (Khan and Altalbe, 2023). Studies have also found that between 50 and 60% of Ukrainians met the clinical diagnostic criteria for post-traumatic stress disorder (PTSD) within the first year of the conflict (Lushchak *et al.*, 2023; Ressler *et al.*, 2024).

Children and youth are among the most vulnerable to the impact of armed conflict (Shenoda *et al.*, 2018; Bendavid *et al.*, 2021). Since the onset of the conflict, young people have experienced significant psychological distress due to exposure to severe trauma, including airstrikes, shelling, and civilian casualties (Skrypnyk and Labenko, 2022). Many have witnessed violence, lost family members, and been displaced from their homes (de Alencar Rodrigues *et al.*, 2022). Research has found conflict affected children and youth in Ukraine have experienced increased externalizing behavioral challenges, as well as internalizing symptoms such as anxiety, depression, and PTSD (Bürgin *et al.*, 2022; McElroy *et al.*, 2024). Studies have found 35% increase in internalizing symptoms among children since the start of the war (McElroy *et al.*, 2024) with approximately 30% of adolescents experiencing moderate to severe depression (Osokina *et al.*, 2023; Goto *et al.*, 2024). Additionally, nearly 70% of children seeking mental health treatment have met the criteria for PTSD (Pfeiffer *et al.*, 2024).

## Conflict and education

The education sector in Ukraine has been severely disrupted since the start of the conflict, with teachers forced to adapt to wartime conditions such as damaged infrastructure, air raid alarms, frequent displacement, and working in unsafe environments (Nenko *et al.*, 2023). Schools and universities have been targeted by Russian missiles, resulting in the damage or destruction of over 3,700 educational institutions, inflicting physical and emotional trauma on teachers, staff, and students (Human Rights Watch, 2023).

Because of these challenges, educators have had to significantly adjust their practices, including modifying schedules, managing increased workloads, conducting remote virtual lessons, providing tutoring, and offering psychological support to students affected by trauma (Orlov, 2022; Velykodna *et al.*, 2023). In many cases, opportunities for professional development have also been halted as training and advancement become scarce or unavailable in war-torn areas.

Teachers have played a pivotal role in supporting children affected by the conflict, yet balancing these demands in addition to coping with their own trauma can place significant strain on their mental health and well-being (Velykodna *et al.*, 2023; Kang *et al.*, 2024; Rzońca *et al.*, 2024). This dual burden can result in a range of psychological symptoms such as stress, anxiety, burnout, depression, and post-traumatic stress disorder (PTSD; Sharifian and Kennedy, 2019). Research indicates that nearly 33% of Ukrainian educators reported high stress levels, while 36% were experiencing burnout symptoms, including disengagement and emotional exhaustion (Velykodna *et al.*, 2023).

Considering these challenges, support services and interventions are needed to buffer the stress experienced by teachers living in conflict affected regions. For instance, strong social support networks can reduce feelings of stress and isolation (Veronese *et al.*, 2018). Additionally, professional development opportunities that emphasize trauma-informed teaching practices and self-care strategies can increase educators' knowledge and skills on managing their own stress and effectively supporting their students (Sharifian and Kennedy, 2019; Sharifian *et al.*, 2023). Furthermore, the implementation of stress management and self-care workshops for teachers can contribute to their overall well-being and ability to cope with the ongoing challenges of teaching in a conflict-affected environment (Sharifian *et al.*, 2023).

## Mental health services in Ukraine

Since the onset of the war, Ukraine's mental health care system has been overwhelmed by the escalating need for services (Seleznova *et al.*, 2023). The conflict between Russia and Ukraine, which began in 2014, prompted international entities such as the World Health Organization (WHO) to launch initiatives aimed at addressing the treatment gap and expanding mental health supports and services in Ukraine (Chaulagain *et al.*, 2020; World Health Organization, 2023). Programs such as the Mental Health Gap Action Program (mhGAP) and Self-Help Plus (SH+) have been implemented to scale-up mental health services in the country (World Health Organization, 2023). Both mhGAP and SH+ seek to expand the knowledge and skills of non-specialized providers to deliver low-intensity psychological interventions to individuals with mental health needs (World Health Organization, 2021; Petagna *et al.*, 2023). mhGAP provides guidelines and tools for non-specialist healthcare providers (e.g., doctors and nurses) to integrate mental health supports into primary care. The program's goal is to equip healthcare workers (e.g., doctors, nurses) with knowledge and skills needed to diagnose and treat mental health conditions, thus improving access to mental health services in primary care settings (Petagna *et al.*, 2023). SH+ is a low-intensity group based psychological intervention, designed to teach stress management skills with the goal of reducing psychological distress and the likelihood of developing future mental health disorders (World Health Organization, 2021). Rooted in principles of cognitive behavioral therapy (CBT), SH+ aims to help individuals learn alternative ways to manage challenging thoughts and feelings (Epping-Jordan *et al.*, 2016).

The intervention includes a pre-recorded course, and a self-help book that is delivered across five two-hour sessions with groups of 20–30 individuals (Purgato et al., 2019).

A Universal Mental Health Training (UMHT) for front-line specialists (police officers, social workers, pharmacists, etc.), which was developed and piloted in Ukraine, is another intervention designed to address the shortage of specialized mental health providers in underserved regions of Ukraine (Gorbunova et al., 2024). The UMHT 5-step model includes training on: (1) recognizing mental health conditions; (2) validating mental health concerns; (3) providing support; (4) referring an individual to professional help; and (5) ensuring an individual receives available mental health resources. The objectives of the UMHT are to improve knowledge, awareness and readiness to support individuals experiencing psychological distress (Gorbunova et al., 2024).

While psychosocial support and training interventions have expanded in Ukraine over the past decade, there are few that are contextualized to the political environment, protracted conflict, and the specific needs of educators in Ukraine. Moreover, most teacher-focused mental health trainings lack robust evidence, do not address teachers mental health and well-being, and are delivered in person (Bogdanov et al., 2017; Anderson et al., 2019). As such, there is an urgent need to test the effectiveness of mental health training interventions tailored for the specific experiences of educators in Ukraine.

The goal of this non-randomized control trial (NRCT) is to examine the effectiveness of the Psychosocial Support for Educators (PSE) program, an adaptation of the UMHT, is designed to meet the unique needs of teachers, specifically those residing in conflict-affected regions of Ukraine. To do this, the primary aims of this study are (1) to test if the PSE improves educators' knowledge and readiness to provide psychosocial support to children and adolescents, and (2) to explore if participation in the PSE training intervention influenced participants' mental health and well-being. We hypothesize that participants of PSE program will report: *H1.* increased knowledge and readiness to provide psychosocial support to children and adolescents; and *H2.* greater reduction in psychological distress symptoms than the control condition.

## Methods

We used an NRCT design, which enabled us to conduct the study in a real-world setting, providing insights into the intervention's effectiveness in a naturalistic environment (Li, 2014). In the context of our study, randomization was not feasible because of conflict-related security concerns such as military shelling, lack of access to electricity and internet, and increased work and personal demands. Data were collected across three time points (pre-intervention, post-intervention, 1-month follow-up) between February and June 2024. Prior to commencing study activities, the protocol was reviewed and approved by the ethical review board from Zhytomyr National University in Ukraine.

### Sample and procedures

Educators were recruited from three regions in Ukraine (Central, Western, and South-Eastern) by the regional departments of education *via* social networks, online fliers and on the Department of Education website. Participants were eligible for the study if they were over the age of 18, held a bachelor's or master's degree, and were presently employed as an education worker in Ukraine. Prior

to enrolling, each participant was provided information about the study and written consent was obtained from those who expressed interest and met the eligibility criteria.

Educators ($N$ = 2,911) were invited to participate and ($n$ = 881) enrolled in the study. Surveys assessing mental health, knowledge, and readiness, were collected at three time points (baseline, post-intervention, and 1-month follow-up). Those who selected the control group were informed that their role would involve completing surveys at three timepoints, whereas those in the experimental group were asked to participate in the training and complete surveys *via* an online questionnaire at three timepoints. A total of 329 participants in the experimental group and 192 participants in the control group completed all survey assessments and were included in the final analyses (Figure 1 visualizes the participant flowchart).

Participants primarily identified as female (96.2%) and were between the ages of 26–44 (44.4%) or 45–59 (44.2%). The majority lived in cities (65.7%) and were geographically distributed across three regions: Central (26.6%), Western (23.7%), and South-Eastern (49.6%). In terms of education, most participants were educated as a specialist (a post-secondary degree positioned between a bachelor's and master's degree), (54.0%), or held a master's degree, (32.7%). Participants were employed in secondary schools (62.4%) or primary schools (26.1%), with the majority reporting 16 or more years of employment in the educational sector (56.8%). The PSE experimental and control group did not significantly differ on demographic characteristics. Regarding war exposure, 15% moved to another region in Ukraine due to the war, 21.5% directly witnessed war related events, and 33.3% were currently separated from their family. The highest level of exposure was reported among participants from the South and East, which are regions that have been most impacted by the war. See Table 1 for demographic characteristics and Table 2 for all war exposure variables separated by region.

### Intervention–psychosocial support for educators

The "Psychosocial Support for Educators" intervention was adapted from the UMHT model (described earlier) by a team of Ukrainian researchers and Smart Osvita, a Ukrainian organization that provides teachers with professional development trainings and workshops on a variety of topics such as mental health, innovative teaching methods, and supporting children with special educational needs (Smart Osvita, 2019). To ensure the intervention was tailored to the Ukrainian educational context, three researchers and four Smart Osvita staff independently reviewed the UMHT model to identify components relevant to educators. The team then engaged in a collaborative, iterative adaptation process – discussing potential additions, reconciling differing perspectives, and refining the intervention to ensure contextual fit and responsiveness to educators needs amid ongoing conflict.

The PSE consists of six modules, with each module following five steps of the UMHT intervention model including: (1) recognizing mental health conditions; (2) validating mental health concerns; (3) providing support; (4) referring an individual to professional help; and (5) ensuring an individual receives available mental health resources. Educator-specific content that was incorporated to the PSE intervention includes: (1) common mental health conditions in trauma and war affected children; (2) basic psychosocial support for children with trauma-related stress responses; (3) healthy coping strategies to reduce stress in children; (4) psychoeducation and support to parents of youth experiencing distress. Additionally,

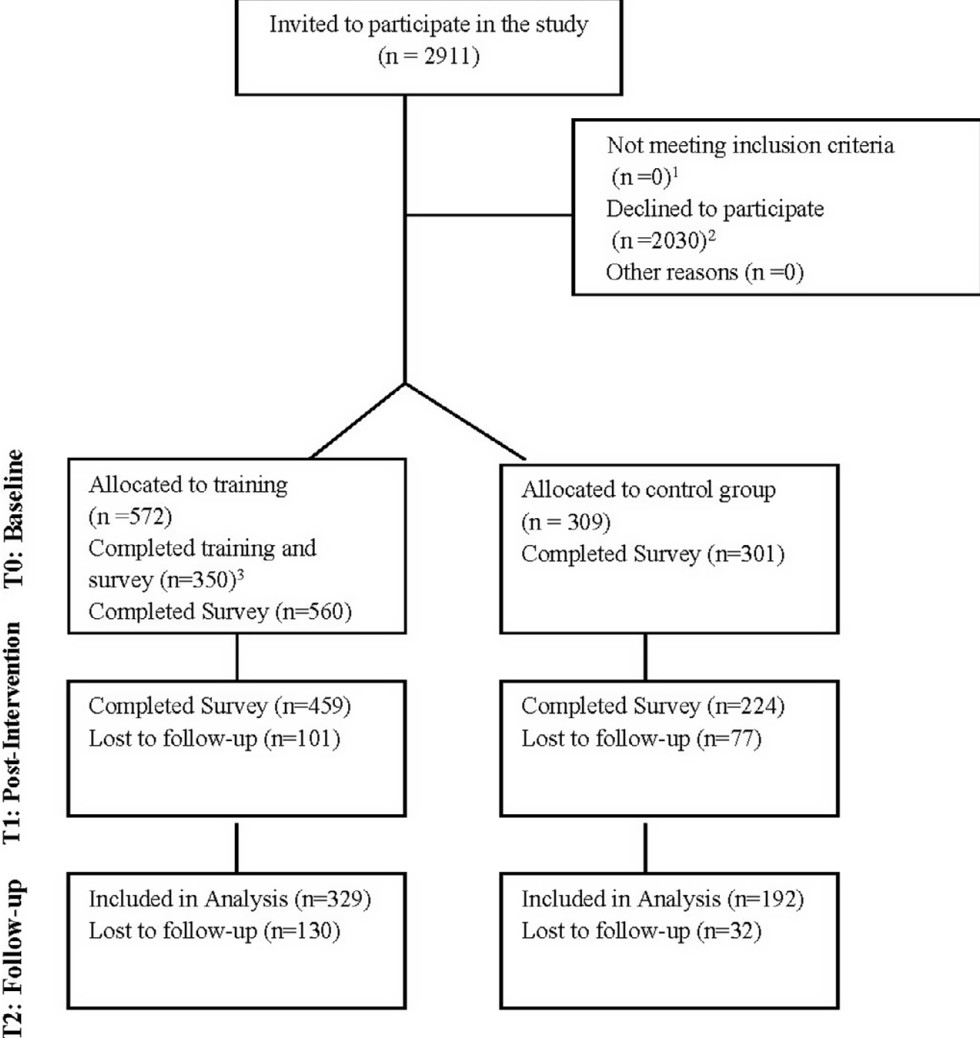

**Figure 1.** Participant flowchart.
[1] All participants who were invited to enroll in the study were educators through the department of education and therefore met the inclusion criteria.
[2] The primary reasons for declining to participate included: high workloads due to the war; no electricity and internet in the school, security concerns related to shelling; and personal issues (illness, or illness of relatives).
[3] This includes participants who fulfilled all training requirements including completion of the curriculum, attending supervision, and completion of survey assessments.

common stress responses among educators, self-care, and mental health supports for teachers were integrated into each module.

The intervention included a combination of synchronous and asynchronous lessons across six modules consisting of live presentations, recorded videos, self-assessment quizzes, and supplemental reading materials. The synchronous sessions were delivered by psychologists through the Open LMS, a learning management system that provides a customizable and flexible environment for delivering online training programs. These sessions were conducted with groups of 15–30 educators over 11 consecutive days with each session lasting three hours. Synchronous sessions were also recorded to ensure accessibility for those facing internet connectivity issues or who are unable to attend. Asynchronous sessions were also available *via* the online platform to ensure accessibility for teachers in remote and rural regions where in-person training was not feasible.

The delivery techniques used in the intervention include psychoeducation, interactive exercises, solution-focused activities, and mutual support. Additionally, ongoing group supervision was provided by the developers of the program to the psychologists who delivered the program for one hour per month (see Table 3 for module content). Supervision addressed program delivery, fidelity to the intervention model, and challenges related to implementation, along with strategies for addressing them.

### Measures

Measures of mental health, knowledge, and readiness were administered across three time points (pre-intervention, post-intervention, and 1-month follow up). All measures of mental health were administered in the Ukrainian language and have been validated and available in Ukraine.

The Mental Health Stability Questionnaire Short Form (MHC-SF-UA) assessed well-being across three dimensions: emotional, psychological, and social. This scale consists of 14 items that respondents rate based on their frequency of experiences related to personal growth, positive relationships, and emotional balance, using a six-point Likert scale from 0 (Never) to 5 (Every day; Lamers *et al.*, 2011). The measure consists of three emotional well-being items (range 0–15), six psychological well-being items (range 0–30),

**Table 1.** Demographic data

| Demographic category | | Total n (%) | PSE group n (%) | Control group n (%) | Chi square $\chi^2$ |
|---|---|---|---|---|---|
| Age | 18–25 | 40 (7.04) | 31 (9.34) | 9 (4.64) | 3.95, p = .27 |
| | 26–44 | 234 (44.4) | 145 (43.67) | 89 (45.88) | |
| | 45–59 | 233 (44.2) | 145 (43.67) | 88 (45.36) | |
| | 60+ | 19 (3.6) | 11 (3.31) | 8 (4.12) | |
| Sex | Male | 20 (3.8) | 13 (3.92) | 7 (3.61) | .99, p = .58 |
| | Female | 506 (96.2) | 319 (96.08) | 187 (96.39) | |
| Region | Central | 140 (26.61) | 97 (29.22) | 43 (22.16) | 5.37, p = .07 |
| | Western | 125 (23.70) | 80 (24.10) | 45 (23.20) | |
| | South-Eastern | 261 (49.62) | 155 (46.69) | 106 (54.64) | |
| Location of home | City | 346 (65.77) | 225 (67.77) | 121 (62.37) | 3.50, p = .17 |
| | Village | 155 (29.47) | 95 (28.61) | 60 (30.93) | |
| | other | 25 (4.75) | 12 (3.61) | 13 (6.70) | |
| Education sector employment | Higher | 11 (2.09) | 7 (2.11) | 4 (2.06) | 4.46, p = .49 |
| | Vocational School[1] | 18 (3.42) | 13 (3.92) | 5 (2.58) | |
| | Secondary[2] | 328 (62.36) | 204 (61.45) | 124 (63.92) | |
| | Preschool | 27 (5.13) | 21 (6.33) | 6 (3.09) | |
| | Extra-Curriculum[3] | 5 (.90) | 4 (1.20) | 1 (0.52) | |
| | Primary[4] | 137 (26.05) | 83 (25.00) | 54 (27.84) | |
| Level of education | Master | 172 (32.7) | 109 (32.83) | 63 (32.47) | 2.47, p = .48 |
| | Bachelor | 18 (3.42) | 13 (3.92) | 5 (2.58) | |
| | Specialist[5] | 284 (54.00) | 174 (52.41) | 110 (56.70) | |
| | Junior bachelor[6] | 52 (9.89) | 36 (10.84) | 16 (8.25) | |
| Years of experience as an educator | 0–5 | 81 (15.4) | 54 (16.27) | 27 (13.92) | 1.65, p = .65 |
| | 6–10 | 52 (9.88) | 30 (9.04) | 22 (11.34) | |
| | 11–15 | 94 (17.87) | 57 (17.17) | 37 (19.07) | |
| | 16 or more | 299 (56.84) | 191 (57.53) | 108 (55.67) | |

*Notes*: 1. An educational institution for youth over the age of 15 providing youth specialized and vocational training. 2. Schools providing education to youth aged 10–17. 3. Afterschool programs. 4. Schools providing education to youth aged 6–10. 5. A post-secondary degree positioned between a bachelor's and master's degree. 6. Equivalent to a two-year post-secondary degree.

and five social well-being items (range 0–25). Sample items include "During the past month, how often do you feel satisfied with life?" and "During the past month, how often do you feel that you had experiences that challenged you to grow and become a better person?" The MHC-SF-UA has been tested in numerous languages yielding strong psychometric properties, including high internal consistency, with Cronbach's alpha values of 0.80 or higher (Lamers *et al.*, 2012, 2011; Machado and Bandeira, 2015; Doré *et al.*, 2017). Cronbach's alpha for this sample was $\alpha$ = .92.

The Perceived Stress Scale (PSS-10), a 10-item instrument measured participants' perceived stress (Cohen *et al.*, 1983). Questions inquire how often respondents find their lives unpredictable, uncontrollable, and overloaded. Items are scored on a five-point Likert scale ranging from 0 (Never) to 4 (Very Often). Example items included "How often have you felt difficulties were piling up so high that you could not overcome them?" and "How often have you felt nervous and stressed?" Total scores range from 0 to 40, with higher scores reflecting higher perceived stress levels. The PSS-10 has been translated into over 25 languages demonstrating high

internal reliability, with Cronbach's alpha values between $\alpha$ = .78 and .90 (Taylor, 2014; Michalski *et al.*, 2021; Chudzicka-Czupała *et al.*, 2023). Cronbach's alpha for this sample was $\alpha$ = .79.

The Burnout Measure, Short Version is a 10-item scale assessing indicators of burnout including physical, mental and emotional exhaustion (Malach-Pines, 2005; Lourel *et al.*, 2008). Each item is scored on a seven-point Likert scale with responses ranging from 1 (Never) to 7 (Always). The measure inquires "When you think about your work overall, how often do you feel the following" with each item providing different response options such as: tired, hopeless, or trapped (Malach-Pines, 2005). Scores range from 10 to 70 with higher scores indicating greater burnout. The measure has been tested across multiple languages consistently demonstrating Cronbach's alpha of over 0.70 (Malach-Pines, 2005), and yielding solid construct validity, correlating well with other burnout measurement tools (Capri, 2013). Cronbach's alpha for this sample was $\alpha$ = .91.

The Patient Health Questionnaire-9 (PHQ-9) is a 9-item measure designed to screen, diagnose, and monitor the severity of

**Table 2.** Exposure variables

| Exposure | North/Central N = 140 | South/Eastern N = 261 | West N = 125 | Total N = 520 |
|---|---|---|---|---|
| | | Region n (%) | | |
| Moved to another region in Ukraine | 12 (7.8) | 65 (25.7) | 3 (2.7) | 80 (15.4) |
| House ruined or destroyed | 7 (4.5) | 40 (15.8) | 4 (1.8) | 49 (9.4) |
| Home close to an active war zone | 10 (6.5) | 65 (25.7) | 2(1.8) | 77 (14.8) |
| Directly witnessed war related events | 20 (13.0) | 80 (31.6) | 12 (10.6) | 112 (21.5) |
| House in occupied territory | 11(7.1) | 67 (26.5) | 1 (0.9) | 79 (15.2) |
| Experienced a serious injury due to the war | 19 (12.3) | 50 (19.8) | 11 (9.7) | 80 (15.4) |
| Currently separated from family | 37 (24.0) | 107 (42.3) | 29 (25.7) | 173 (33.3) |

depression. The questions are based on the 9 criteria of depression outlined in the Diagnostic Statistical Manual-IV (DSM-IV) (Kroenke *et al.*, 2001). Each item is scored on a 4-point Likert scale inquiring how often a person experiences different symptoms of depression with responses ranging from 0 = Not at All to 3 = Nearly Every Day. Scores range from 0 to 27 indicating no depression symptoms (0–4), minimal depression symptoms (5–9), mild depression symptoms (10–14), moderately severe depression symptoms (15–19), and severe depression symptoms (20–27). The cut-off score for probable depression is ≥10 (Manea *et al.*, 2015) The PHQ-9 has illustrated reliability and validity across numerous populations and languages with internal consistency, with Cronbach's alpha ranging from 0.71 to 0.88 (Wang *et al.*, 2014; Dadfar *et al.*, 2018; Molebatsi *et al.*, 2020; Sun *et al.*, 2022). Cronbach's alpha for this sample was $\alpha$ = .87.

The Generalized Anxiety Disorder-7 (GAD-7), a 7-item questionnaire assesses generalized anxiety and the severity of symptoms (Spitzer *et al.*, 2006; Aleksina *et al.*, 2024). The questionnaire asks how often, of the period of the last 2 weeks, the individual was bothered by different anxiety symptoms such as: "feeling nervous, anxious, or on edge"; "worrying too much about different things." Each item is scored on a four-point Likert scale ranging from 0 = Not at all to 3 = Nearly every day. Scores range from 0 to 21 with scores of 0–4 indicating minimal anxiety, 5–9 mild anxiety, 10–14 moderate level of anxiety, and 15–21 severe anxiety (Gorbunova *et al.*, 2024). Cronbach's alpha for this sample was $\alpha$ = .89.

The "Understanding the Context of Psychosocial Support" knowledge questionnaire was developed by the research team to assess changes in participants' knowledge of mental health and psychosocial support for students and their parents. The measure consists of 7 items, with questions assessing knowledge in areas such as: (1) signs and symptoms of mental health conditions; (2) mental health stigma; and (3) psychosocial support strategies for children in crisis. Correct items were summed with score ranges from 0 to 21, with higher scores reflecting a higher level of knowledge.

The readiness questionnaire, developed by the research team, assessed perceived readiness to provide basic psychological support and readiness to apply the 5-steps in the PSE intervention. Seven questions were asked on a 5-point Likert scale ranging from 1 (not

at all ready) to 5 (absolutely ready). Sample questions inquired about readiness to: (1) support students and families who are experiencing mental health challenges; (2) provide mental health support to students and families; (3) provide students and families with professional resources to treat mental health conditions; and (4) apply basic psychological support with students and families experiencing distress.

The exposure questions include 10 items, developed by the research team, based on experiences that Ukrainians have encountered since the start of the full-scale invasion. Respondents mark yes or no to each question and each item is scored as no = 0, yes = 1. Items inquire if the person has experienced the following since the start of the 2022 full-scale invasion: (1) changed residences; (2) moved regions; (3) was displaced; (4) house was destroyed; (5) lived in an active war zone; (6) witnessed war related events; (6) felt safe in their home; (7) lived in an occupied territory; (8) was seriously injured during the conflict; (9) separated from their family; (10) took an active role in the conflict.

### *Statistical analyses*

All analyses were performed using SPSS version 29 software (IBM Corp. 2023). Descriptive analyses were carried out using frequencies and percentages. Differences in baseline demographics between PSE and control groups were assessed using Pearson's chi-squared tests.

A linear mixed model (LMM) assessed the effects of the PSE intervention compared to the control group on outcome measures (mental health, well-being, knowledge, and readiness) over time. A LMM was used because of the multilevel structure of the data and to account for variability within and between participants at baseline and across time (Meteyard and Davies, 2020). The fixed effects included Time and Group (PSE vs. Control) with random intercepts for subjects and a random slope for time. To control for potential confounders, we included age, gender, region, length of time working as an educator, and exposure variables including separation from family, if the participants live in an active war zone, and if they were displaced from their home due to the war. The control variables were selected based on prior research indicating their potential influence on mental health outcomes (Eshel *et al.*, 2023; Kimhi *et al.*, 2023). Separate analyses were conducted for each outcome variable.

We evaluated multiple models to determine the best-fitting model for the data. Exploratory data analyses, including examination of variance–covariance matrices and intraclass correlation coefficients (ICCs), supported the assumption of equal correlations among repeated measures. The Akaike Information Criterion (AIC) was used to compare model fit. The best-fitting model which had the lowest AIC and was selected for the final analysis included random intercepts, fixed effects for time and intervention. The covariance structure of the repeated measures was specified as compound symmetry, which assumes equal-sized correlations between measurement occasions (West *et al.*, 2022). This assumption was deemed appropriate for our data as preliminary analyses showed homogeneity in correlation patterns across time points. Restricted maximum likelihood (REML) was used because it yields robust significance tests across varying sample sizes and provides unbiased estimates of random effects in hierarchical data (Meteyard and Davies, 2020). We calculated Cohen's *d* to assess effect size, dividing the interaction estimate (mean difference between groups over time) by the standard deviation of the model's covariance structure. Effect

**Table 3.** Module content

| Modules | Content | Duration (h) |
|---|---|---|
| 1. Mental health of children and adolescents | • Introduction (overview of 5-step model)<br>• Psychoeducation on mental health and mental disorders<br>• Statistics of mental disorders of children and adolescents (Ukraine and world trends)<br>• Mental health support needs and barriers to receiving help<br>• Stigma of in mental health of children and adolescents<br>• Skills to support individuals in distress | 3 |
| 2. Supporting children and adolescents with mental health challenges | • Signs of distress in children and adolescents<br>• Communicating about mental health with children and adolescents across developmental levels<br>• Supporting children and adolescents with mental health difficulties<br>• Strategies for a child/adolescent to support their own mental health<br>• Referring children and adolescents to professional help<br>• Establishing routines to support children's mental health<br>• Integrating self-care into daily routines | 3 |
| 3. Supporting children and adolescents in crisis | • Signs of crisis in children and adolescents<br>• Common reactions to crisis in children and adolescents<br>• Techniques to support young people in crisis<br>• Providing psychoeducation to children and adolescents<br>• Referral of children and adolescents in crisis to professional help<br>• How to ensure a young person receives professional help | 3 |
| 4. Interacting with parents of children experiencing distress | • Common learning and peer relationship challenges in distressed children and adolescents<br>• Engaging parents/caregivers in their child's mental health intervention<br>• How to talk to parents/caregivers about a child's mental health<br>• Strategies to support parents/caregivers of children with mental health difficulties<br>• Referral strategies for parents/caregivers to seek out and/or receive professional help | 3 |
| 5. Stress and resilience in education | • Recognize the signs of and common responses to acute and chronic stress in young people<br>• How to talk to young people about stress and its impact on mental health<br>• Healthy ways teachers can cope with stress<br>• Strategies to support children experiencing stress<br>• Referral strategies for children and parents to professional help | 3 |
| 6. Mental health of children and adolescents during war | • Children's emotional reactions to war (shelling, bomb shelters, loss, evacuation)<br>• How to discuss common mental health reactions to war<br>• Basics of psychological first aid<br>• Supports for children and adolescents during the war<br>• Organizing time in a bomb shelter<br>• Referral for professional help<br>• Home assignments during the war time | 3 |
| Group supervision | Discussion of cases that teachers applied knowledge and skills gained during the training and supervisory support | 1 h/month |

sizes are interpreted as small $d = 0.2$, medium, $d = 0.5$, and large $d = 0.8$ or greater (Cohen, 2013).

## Results

At post-intervention, the PSE group demonstrated significantly greater increases in knowledge compared to the control group ($t = 2.97$, $p = .003$, $d = .38$), with these gains sustained at the 1-month follow-up ($t = 3.03$, $p = .003$, $d = .39$). Additionally, the PSE group showed significantly higher readiness to provide psychosocial support over time compared to the control group, with large effects observed at post-intervention ($t = 6.63$, $p < .001$, $d = .85$) and maintained at follow-up ($t = 4.65$, $p < .001$, $d = .60$). Cohen's $d$ values indicated small to medium effects for knowledge gains and medium to large effects for readiness.

The PSE group also reported greater improvements in social ($t = 4.20$, $p < .001$, $d = .54$), emotional ($t = 2.60$, $p < .01$, $d = .33$), and psychological well-being ($t = 3.72$, $p < .001$, $d = .48$) compared to the control group at post-intervention. These enhancements persisted at 1-month follow-up, with medium to large effect sizes ($d = .50$ to .70).

On the GAD anxiety measure, the PSE group experienced a greater reduction in anxiety compared to the control group at post-intervention ($t = 3.20$, $p = .001$, $d = .41$), with similar small to medium effect sizes observed at follow-up ($t = 3.29$, $p = .001$, $d = .42$). Similarly, the PSE group showed significantly greater reductions on the Perceived Stress Scale (PSS) at post-intervention ($t = 2.70$, $p < .01$, $d = .35$), with additional reductions at 1-month follow-up ($t = 3.59$, $p < .001$, $d = .46$). In terms of depression, the PSE group reported significant reductions across both time points, though these changes reflected smaller effect sizes ($d = .25$–.26).

Finally, the PSE group exhibited greater reductions in burnout compared to the control group at post-intervention ($t = 2.01$, $p < .05$, $d = .26$), with further decreases observed at the 1-month follow-up ($t = 3.18$, $p < .01$, $d = .41$). These results correspond to small to medium effect sizes (Table 4).

## Discussion

This study examined the effectiveness of the virtual PSE program for teachers living in conflict-affected regions of Ukraine. We found that educators who participated in the PSE group reported

**Table 4.** Primary outcomes across time

| | Mean (SE) | | | | | | Group-time interaction | | | | | | |
|---|---|---|---|---|---|---|---|---|---|---|---|---|---|
| | PSE group | | | Control group | | | Pre to post | | | | Pre to follow-up | | |
| Outcome | $t_1$ | $t_2$ | $t_3$ | $t_1$ | $t_2$ | $t_3$ | F | t | d | CI | t | d | CI |
| Knowledge | 14.65 (.63) | 15.53(.63) | 15.59 (.63) | 14.79 (.65) | 14.86 (.65) | 14.92 (.65) | 5.94** | 2.97** | 0.38 | .27, 1.36 | 3.03** | 0.39 | .28, 1.37 |
| Readiness | 3.58 (.13) | 4.17 (.13) | 4.20 (.13) | 3.49 (.14) | 3.54 (.14) | 3.70 (.14) | 23.14*** | 6.63*** | 0.85 | .37, .75 | 4.65*** | 0.6 | .21, .52 |
| Burnout | 21.14 (1.98) | 18.85(1.98) | 17.20 (1.98) | 22.38 (2.03) | 23.01 (2.03) | 23.69 (2.03) | 5.16** | 2.01* | 0.26 | .03, 3.19 | 3.18** | 0.41 | .98, 4.14 |
| PHQ | 6.41 (.86) | 5.18(.87) | 4.71(.86) | 7.21(.89) | 6.73(.89) | 6.25(.89) | 2.61$^†$ | 2.00* | 0.26 | .01, 1.5 | 1.96* | 0.25 | .03, 1.47 |
| GAD | 6.94 (.84) | 5.71 (.85) | 5.07 (.85) | 7.47 (.86) | 7.42 (.87) | 6.81 (.87) | 7.03*** | 3.20** | 0.41 | .45, 1.88 | 3.29** | 0.42 | .48, 1.91 |
| PSS–10 | 18.80(.96) | 16.73(.96) | 15.82(.96) | 19.39 (.99) | 18.54(.99) | 18.02(.99) | 6.98*** | 2.70** | 0.35 | .33, 2.10 | 3.59*** | 0.46 | .73, 2.49 |
| Social well-being | 16.15 (.87) | 17.49(.87) | 17.73(.87) | 15.77(.90) | 15.35(.90) | 15.38(.90) | 13.25*** | 4.20*** | 0.54 | .94, 2.57 | 4.68*** | 0.6 | 1.13, 2.78 |
| Emotional well-being | 10.25(.60) | 11.02 (.60) | 11.52 (.60) | 9.62(.61) | 9.74(.61) | 9.83(.61) | 9.12*** | 2.60** | 0.33 | .15, 1.14 | 4.23*** | 0.54 | .566. 1.55 |
| Psychological well-Being | 21.79 (1.05) | 23.10(1.05) | 23.79(1.05) | 21.42 (1.08) | 20.96 (1.08) | 21.16 (1.07) | 12.48*** | 3.72*** | 0.48 | .84, 2.72 | 4.75*** | 0.73 | 1.33, 3.21 |

$^†p < .1, *p < .05, **p < .01, ***p < .001.$

significantly greater improvements in mental health, well-being, knowledge, and readiness to provide psychological support compared to a control condition. This is, to our knowledge, among the one of the first studies to examine a virtual educator-focused psychosocial intervention in a conflict-affected region.

One of the primary objectives of the PSE intervention was to enhance educators' knowledge and readiness to provide psychosocial support to children and adolescents. Educators in the PSE group showed significant improvements compared to the control group in their knowledge of mental health and psychosocial support strategies, with small to medium effect sizes observed at both post-intervention and follow-up. This finding aligns with other research examining knowledge acquisition in educator mental health training programs (Ohrt *et al.*, 2020; Liao *et al.*, 2023). The importance of knowledge of mental health and psychosocial support cannot be overstated in the context of conflict-affected regions. As noted in a systematic review by Burde *et al.*, (2015), teacher-focused interventions that enhance knowledge of mental health and psychosocial support are crucial in conflict-affected regions, to equip teachers to provide appropriate psychological support to their students. The sustained improvement in knowledge observed in our study suggests that the PSE training intervention may increase teacher preparation for dealing with the complex psychosocial needs of students in conflict-affected areas.

Those who participated in the PSE intervention also reported significant increases in readiness to provide psychosocial support, with medium to large effect sizes. Research has shown that when teachers learn more about mental health concepts, how to recognize distress, and ways to provide support, they become more capable of applying this knowledge in their work (Manjari and Srivastava, 2020). Increasing teachers' understanding of mental health also helps to reduce barriers like stigma and misconceptions, further boosting their readiness to engage in supportive practices (Maclean and Law, 2022). Moreover, studies have found that as teachers gain knowledge about mental health concepts, signs of distress, and support techniques, they become better prepared to provide supportive care to their students (Falk *et al.*, 2019). The findings indicate that the PSE may contribute to positive outcomes such

as these by improving educator capacity for delivering psychosocial support.

A secondary objective of the study was to explore if participation in the PSE intervention influenced participants' mental health and well-being. Significantly greater reductions in stress and positive increases in social, emotional, and psychological well-being were observed at both post-intervention and follow-up compared to the control group. Results also illustrated significant reductions in symptoms of anxiety, depression, and burnout that were sustained through 1-month follow up. These findings are encouraging due to the high rates of psychological distress such as burnout, depression, and anxiety that Ukrainian teachers have reported since the start of the conflict (Velykodna *et al.*, 2023). Compromised teacher well-being often results in high attrition, impacts professional and student performance, and can negatively influence teacher-student relationships (Carver-Thomas and Darling-Hammond, 2017; Falk *et al.*, 2022). These relationships are fundamental to creating a supportive and nurturing learning environment, which is particularly essential for students in conflict-affected regions who may be dealing with trauma and stress (Falk *et al.*, 2019, 2022). When teachers are overwhelmed or struggling with their own well-being, their ability to form positive, supportive relationships with students can be impaired (Velykodna *et al.*, 2023). This can result in a less supportive classroom environment and reduced emotional support for students who may desperately need it (Sharifian and Kennedy, 2019). The positive effects observed in this study suggest that interventions such as the PSE can serve as a protective factor, potentially mitigating these negative outcomes. By improving teachers' well-being across multiple domains, such programs may help in supporting educators' mental health, thereby reducing attrition rates, maintaining or improving student achievement levels, and fostering more positive teacher-student relationships. This, in turn, could contribute to a more supportive educational environments in the face of ongoing challenges. Future research could further explore the long-term impacts of such interventions on educator mental health, teacher retention rates, student outcomes, and the quality of teacher-student relationships in conflict-affected settings. Additionally, investigating how improvements in teacher well-being translate into classroom practices and school

climate could provide valuable insights for developing comprehensive support systems for educators in challenging environments.

Surprisingly, trauma exposure among educators participating in the intervention was relatively low. As expected, the highest rates of exposure were reported by participants from the southeastern region of Ukraine, where 42% were separated from their families, 30% directly witnessed war-related events, and 26% were living in occupied territories. Although these rates reflect higher trauma exposure compared to educators in other regions, they are lower than anticipated given the severity of the war's impact in southeastern Ukraine. One possible explanation is that educators who experienced more severe trauma – such as displacement or the loss of their homes – may have been unable to participate in the intervention due to ongoing instability, limited access to technology, or heightened psychological distress. This highlights the importance of considering barriers to participation among those most affected by conflict.

Recruiting and retaining participants was a notable challenge, a foreseeable difficulty considering the intervention took place during an active conflict. Educators identified lack of time as a primary obstacle to participation in the study, as many lived in regions where the ongoing invasion made it unfeasible to take part in the training. Connectivity issues were also a significant barrier, with many reporting lack of electricity and unstable internet connection, which suggests that future trainings might consider alternative delivery strategies, such as in-person or asynchronous, especially if infrastructure challenges worsen. Additionally, varying levels of digital literacy among participants posed challenges in utilizing online resources and tools, highlighting the highlighting the need for user-friendly platforms and technical support, as well as plans for digital engagement. The intensity of the training – four-hour daily sessions in addition to participants' regular workload – further limited consistent attendance and the application of acquired knowledge. Future implementation could of the intervention could include shorter, more flexible modules, asynchronous components, or extended timelines. These changes may better align with educators' demanding schedules and support integration of training content into their professional roles.

These challenges underscore the complex realities of implementing educational interventions in conflict-affected settings and the need for flexible, context-sensitive approaches. Additional strategies to improve retention could include scheduling training sessions during holiday breaks to prevent added workload for educators, engaging educators in Smart Osvita events beforehand to build trust and familiarity with the organization and offering compensation for participants' involvement in the research.

## Limitations

There were several limitations and challenges in this study that are typical of working in conflict zones. First, due to the nature of the conflict, we were unable to employ a randomized control trial. The self-selection process for the PSE program may have resulted in a more motivated intervention group, resulting in difficulties in recruiting and retaining participants for the control group across all three stages of questionnaire completion.

To address the inherent limitations of a non-randomized design, we employed a linear mixed model (LMM) (Bono *et al.*, 2021). The use of LMM allowed us to account for potential baseline differences between groups by incorporating covariates, helping to adjust for

pre-existing variations that could influence the outcomes (Jost and Jansen, 2022). Furthermore, given the repeated measures nature of our study, the LMM was essential in accommodating within-subject variability over time, allowing for a more flexible covariance structure (Meteyard and Davies, 2020; Muradoglu *et al.*, 2023). This approach helped to control for individual differences and provided a more accurate estimation of the intervention's effects (Bono *et al.*, 2021). By incorporating both fixed and random effects, we were also able to adjust for confounding variables, mitigating biases associated with the non-randomized design (Kliegl *et al.*, 2011; Muradoglu *et al.*, 2023).

The use of self-report measures provided important insights on educator's experience with the PSE program yet is subject to certain limitations. Self-reports rely on participants' subjective perceptions and may not always accurately reflect objective changes in behavior or mental health. This limitation could be addressed in future research by incorporating more objective measures or observational data to complement self-report findings.

## Conclusion

This study is among the first to examine the effectiveness of a virtually available and contextually modified mental health training program for educators in conflict-affected Ukraine. Virtual interventions such as the PSE may increase the overall accessibility of psychological support services and trainings, particularly in areas where in-person service delivery may not be attainable due to conflict-related or other factors that negatively impact local resources (Ruzek *et al.*, 2016; Danese *et al.*, 2025). The findings underscore the value of virtual interventions in increasing accessibility to support services when in-person delivery is not feasible. By supporting educators' well-being and ability to support students, the PSE program adds to the growing body of research on psychosocial support and training initiatives aimed at enhancing educators' capacity to assist trauma-affected children.

**Open peer review.** To view the open peer review materials for this article, please visit http://doi.org/10.1017/gmh.2025.10014.

**Supplementary material.** The supplementary material for this article can be found at http://doi.org/10.1017/gmh.2025.10014.

**Data availability statement.** The data that support the findings of this study are available from the corresponding author, Powell, T., upon reasonable request.

**Acknowledgements.** Americares, Smart Osvitas and The Cencora Impact Foundation.

**Author contribution.** P.T.: C&D, DA&I, W&D, S&O, AFM; P.N.: C&D, DC, DA&I, W&D, AFM; T.I.: C&D, DC, DA&I, W&D, AFM; S.O.: C&D, DC, W&D, FA, AFM; S.T.: C&D, W&D, AFM; M.O.: C&D, DC, S&O, AFM; C.K.: C&D, DC, AFM; M.J.M.: W&D, AFM and C.R.: C&D, W&D, FA, AFM Conceptualization and Design (C&D), Data Collection and Processing (DC), Data Analysis and Interpretation (DA&I), Writing and Drafting (W&D), Funding Acquisition (FA), Supervision and Oversight (S&O), Approval of Final Manuscript (AFM), Project Administration (PA).

**Financial support.** The Cencora Impact Foundation.

**Competing intersts.** M.O. and C.K. work for Smart Osvitas. C.R. worked for Americares.

**Ethics statement.** The study was reviewed and approved by the ethical review board from Zhytomyr National University in Ukraine.

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
