## [Reviewer Report]

The article will be a great addition to the emerging pool of mental health data from Ukraine. It is concisely written, and the study itself is well-planned and clearly describes its limitations.

---

## [Reviewer Report]

Overall, this is a well-written and timely research study that tests a novel digital psychosocial intervention for educators during active conflict. I took note of the researchers ability to recruit a large sample of participants using various passive recruitment strategies as well as later challenges with retention and completion. The results are notable in terms of improvements in teachers ability to support the needs of children and adolescents as well as improvements to their own mental health and wellbeing. Below, I have indicated some areas for minor improvement to further strengthen and clarify the manuscript.

Impact statement:

1. Specify Ukraine in the first sentence of the impact statement for clarity.

Educators are amongst those significantly affected by the conflict,

Background

2. Please clarify an increase compared to when?

Studies report a 35% increase in internalizing symptoms among children (McElroy et al. 2024) with approximately 30% of adolescents experiencing moderate to severe depression (Goto et al. 2024; Osokina et al. 2023).

3. Was the objective to test an existing psychosocial intervention for educators or did the authors develop the intervention under study? Please clarify these details in the last part of the background section where study aims are reviewed.

4. Please engage challenges with retention and completion. Currently, the authors include this as part of the limitations section but I believe that this should be included in the discussion and reflected upon a main finding of the study, particularly given the limited availability of such interventions for active conflict settings. What do these findings suggest and how might they be strengthened to avoid drop out? In addition to the suggestions for later adjustments that the authors note (e.g., in person, etc.), they may also consider the importance of recruitment and retention strategies (i.e., implementation strategies) that could support ongoing participation.

5. More details about the intervention would be helpful. For example:

I’m not clear if this intervention existed previously and/or was adapted to the Ukrainian context. What process was undertaken to adapt/translate the model to Ukraine educators? Who participated in that process? Conducted over 11 consecutive days? How long was each day’s material? Who facilitated the synchronous sessions?

6. Group supervision is provided to who? How were the facilitators identified/selected? What was the nature and frequency of the supervision? What is meant by the training psychologist?

7. Given that the intervention occurred during the war, I was somewhat surprised by relatively lower rates of trauma exposure among the teachers. I was expecting this to be higher. Can the authors engage with this vis a vis the results and suitability for other active conflict settings.

---

## [Reviewer Report]

The authors have fully addressed all areas identified for revision and clarification. I have not further revisions to suggest. This paper will make an important contribution to the literature on mental health interventions in contexts of active conflict.